# Overexpression of Barley Transcription Factor *HvERF2.11* in *Arabidopsis* Enhances Plant Waterlogging Tolerance

**DOI:** 10.3390/ijms21061982

**Published:** 2020-03-13

**Authors:** Haiye Luan, Baojian Guo, Huiquan Shen, Yuhan Pan, Yi Hong, Chao Lv, Rugen Xu

**Affiliations:** 1Key Laboratory of Plant Functional Genomics of the Ministry of Education, Yangzhou 225009, China; luanhaiye@163.com (H.L.); dx120180077@yzu.edu.cn (Y.P.); 2Jiangsu Key Laboratory of Crop Genomics and Molecular Breeding, Yangzhou 225009, China; bjguo@yzu.edu.cn; 3Co-Innovation Center for Modern Production Technology of Grain Crops, College of Agriculture, Yangzhou 225009, China; jsycshq@163.com; 4Institutes of Agricultural Science and Technology Development, Yangzhou University, Yangzhou 225009, China; HongYi691X@163.com; 5Institute of Agricultural Science in Jiangsu Coastal Areas, Yancheng 224002, China; lvchao726240@126.com; 6Institutes of Agricultural Science and Technology Development, Yangzhou University, Wenhui East Road NO. 48, Yangzhou, Jiangsu 225009, China

**Keywords:** barley, ERF transcription factors, waterlogging stress, transgenic *Arabidopsis*, reactive oxygen species

## Abstract

Waterlogging stress significantly affects the growth, development, and productivity of crop plants. However, manipulation of gene expression to enhance waterlogging tolerance is very limited. In this study, we identified an ethylene-responsive factor from barley, which was strongly induced by waterlogging stress. This transcription factor named *HvERF2.11* was 1158 bp in length and encoded 385 amino acids, and mainly expressed in the adventitious root and seminal root. Overexpression of *HvERF2.11* in *Arabidopsis* led to enhanced tolerance to waterlogging stress. Further analysis of the transgenic plants showed that the expression of *AtSOD1*, *AtPOD1* and *AtACO1* increased rapidly, while the same genes did not do so in non-transgenic plants, under waterlogging stress. Activities of antioxidant enzymes and alcohol dehydrogenase (ADH) were also significantly higher in the transgenic plants than in the non-transgenic plants under waterlogging stress. Therefore, these results indicate that *HvERF2.11* plays a positive regulatory role in plant waterlogging tolerance through regulation of waterlogging-related genes, improving antioxidant and ADH enzymes activities.

## 1. Introduction

Waterlogging is a serious abiotic stress affecting crop production. Plants subjected to waterlogging lack oxygen and produce less adenosine triphosphate (ATP), thereby inhibiting root growth which in turn inhibits plant growth and development [1]. In order to cope with waterlogging stress plants adapt various responses such as using glycolysis and fermentation to supply ATP [2], or shifting their energy metabolism from aerobic to anaerobic [3]. The responses also involve the change of plant hormones, gene expression and enzyme activities. For example, while ethylene, abscisic acid (ABA) and gibberellic acid (GA) increase [4,5,6], cytokinin (CK) and auxin (IAA) reduce [7,8]. Similarly, genes encoding enzymes involved in the glycolysis/gluconeogenesis pathway or associated with sucrose metabolism such as pyruvate decarboxylase (PDC) gene and alcohol dehydrogenase (ADH) gene were most upregulated, while genes encoding invertase or encoding chlorophyll a-b binding protein involved in the light-harvesting complex of photosystem II (PSII) were downregulated under waterlogging stress [9,10]. Many stress-associated transcription factors (TFs) were also found to vary in expression level under waterlogging stress [11,12,13]. For example, ERFs and WRKYs were upregulated, while DNA-binding with one finger (Dof) TFs were down-regulated under waterlogging stress [13].

ERFs belong to the transcription factor family APETALA2/ERF, which is one of the largest families of TFs in plants. All the members in this family share an AP2/ERF domain consisting of approximately 60 to 70 amino acids [14]. On the basis of the number of conserved AP2/ERF domain and the gene function, the AP2/ERF TFs can be divided into three subfamilies. They are AP2 (containing two AP2/ERF domains), RAV (containing one AP2/ERF domain and one B3 domain) and ERF (containing a single AP2/ERF domain) [15,16]. In *Arabidopsis* the ERF subfamily can be further divided into 12 subgroups covering 163 genes while in rice it can be further divided into 15 subgroups covering 144 genes [16]. The subgroup VII of the ERF subfamily in *Arabidopsis* and rice has been shown to play an important role in the adaptation to hypoxic conditions [17,18,19,20]. *HYPOXIA RESPONSIVE ERF1* (*HRE1*), *HRE2*, *RELATED TO APETALA2.12* (*RAP2.12*), *RAP2.2*, *RAP2.3*, *SUB1A* and *SNORKEL1/2* are members of the ERF-VII group, which contain a conserved N-terminal motif (MCGGAI/L, termed the MC motif) and are essential for low-oxygen signaling [18,19,20,21,22,23]. *HRE1* and *HRE2* improved the tolerance of low-oxygen stress by up-regulated genes related to anaerobic and ethanolic fermentation such as *PDC1* and *ADH1* [20]. Overexpression of *RAP2.12* and *RAP2.2* in *Arabidopsis* also significantly improves plant survival rate by inducing sugar metabolism, fermentation and ethylene biosynthesis genes [18,20]. *RAP2.3* modulates oxidative and osmotic stress tolerance under submerged conditions [23]. In rice overexpression of *SNORKEL1* and *SNORKEL2* improves flooding tolerance and promotes the internode elongation under submerged conditions [24], while *SUB1A* confers submergence tolerance by reducing elongation growth and consumption of carbohydrates [21,25]. It appeared that most of the subgroup VII members are regulated by the N-end rule pathway, which senses oxygen under waterlogging stress [19].

Barley (*Hordeum vulgare* L) is generally sensitive to waterlogging, which results in a 20%-30% irreversible yield loss [26]. Our previous study identified that one doubled haploid (DH) line, TF58, was waterlogging tolerant and another DH line, TF57, was waterlogging intolerant [27]. To see if there was different in the expression of ERFs between the two DH lines, mRNAs in the plants were sequenced by the next-generation sequencing technology. Analysis of the sequencing data revealed that an ERF sequence was significantly more abundant in TF58 than in TF57. In this study, we isolated the full-length ERF gene from barley, analyzed its sequence feature and investigated its expression profile and function under waterlogging stress. What we found was that this ERF gene named *HvERF2.11* contained a motif (MCGGAI/L) at the N terminus, was highly induced in the waterlogging tolerance line under waterlogging stress, and significantly enhanced waterlogging tolerance in transgenic *Arabidopsis*. Further analysis of the transgenic plants discovered that *AtSOD1*, *AtPOD1* and *AtACO1* increased rapidly under waterlogging stress. Our study provides an insight into the role of ERF genes in barley waterlogging tolerance.

## 2. Results

### 2.1. Sequence Feature of HvERF2.11

We previously genome-widely analyzed APETALA2/ethylene-responsive factor (AP2/ERF) gene family in barley [28] and noticed that a RNA-Seq sequence from an APETALA2/ethylene-responsive factor named *HvERF2.11* was much more abundant in TF58 plants treated with waterlogging than in the same plants treated with the normal irrigation condition. This finding led us to further study this gene. We thus followed a standard procedure to isolate *HvERF2.11* from TF58 roots. Sequencing of *HvERF2.11* showed that the full-length gene was 1158 bp in length and encoded 385 amino acids with a predicted molecular weight of approximately 41.38 kDa and an isoelectric point (PI) of 5.14 (Figure 1A). Aligning its amino acid sequence to the barley sequence database revealed that *HvERF2.11* contained a conserved N-terminal motif (MCGGAI/L) and an AP2 domain at positions of amino acid 123-184. It is known that the N-terminal motif designated as the CMVII-1 motif is the characteristic feature of the ERF-VII group [16]. Accordingly, *HvERF2.11* could be classified as a member of the ERF-VII group.

BLAST search showed that *HvERF2.11* shared a high sequence identity with its homologs from *Triticum aestivum* (89%), *Aegilops tauschii* (88%) and *Brachypodium distachyon* (66%) (Figure 1B). To investigate its evolutionary relationship with its homologs from other plant species, we performed phylogenetic analysis as shown in Figure 1C. *HvERF2.11* shares its sequence homology with its counterparts from other plant species, but is closer to the homologs from *Aegilops tauschii* and *Triticum aestivum*. Of the species compared, three groups can be classified according to the phylogenetic tree (Figure 1C). *HvERF2.11* and the homologs from *Aegilops tauschii* and *Triticum aestivum* belong to the same group, the homologs from *Setaria italic*, *Oryza sativa* and *Sorghum bicolor* go to another group, while the homologs from *Brachypodium distachyon* and *Zea mays* fall into the third group. It is of notice that there are 11 single nucleotide polymorphisms (SNPs) across the whole gene region between TF57 and TF58 (Figure 2A), of which six SNPs alter amino acids in the protein sequence (Figure 2B). Especially, one changed amino acid occurred within the AP2 domain at the position 176 in relative to the sequence from TF58. It was R in TF57, while was K in TF58. Whether the nucleotide or amino acid differences are the cause to lead to the difference of waterlogging tolerance between TF57 and TF58 is yet unknown.

BLAST search also revealed that 5 ERF members from *Arabidopsis*, 19 from rice and 13 from barley, which contained the N-terminal motif (Appendix A). A phylogenetic analysis with the entire protein sequences including (MCGGAI/L) of the *Arabidopsis*, rice and barley showed that *HvERF2.11* is the closest to AK357064 and Os09g0434500. By contrast, it is distant from *RAP2.12* and *RAP2.2*, and more distant from *HRE1* and *HRE2*. *RAP2.12* and *RAP2.2* are clustered into one group, while *HRE1* and *HRE2* are clustered into another group. *RAP2.12*, *RAP2.2*, *HRE1* and *HRE2* have been shown to function in the regulation of the hypoxia stress response [22,23]. Rice *SNORKEL1* and *SNORKEL2* are clustered into one group as expected, while *SUB1A*, *SUB1B*, *SUB1C* and *Os09g0287000* form another cluster. In particular, these genes are distant from the genes related to waterlogging in *Arabidopsis* (Figure 3). Considering that rice is monocot while *Arabidopsis* is dicot, these results suggest that monocots and dicots might have different waterlogging-tolerant mechanisms.

### 2.2. Expression of the HvERF2.11 Gene under Waterlogging Stress

The expression of *HvERF2.11* in different tissues of two DH lines TF57 and TF58 under the normal and waterlogging conditions was quantified using qRT-PCR. *HvERF2.11* was demonstrated to express differently in different tissues of each line under either the normal condition or the waterlogging condition (Figure 4A,B). It was highly expressed in adventitious roots and seminal roots, but lowly expressed especially in leaves, in both lines. The highest expression level from the seminal roots is about 9.0-fold the lowest expression level from the leaves in either TF57 or TF58 under the normal condition. In addition, *HvERF2.11* showed a differential inducible pattern between TF57 and TF58. In TF58 *HvERF2.11* expressed at a higher level in adventitious roots, seminal roots and nodal roots than in TF57 under waterlogging stress. In more details, the level of *HvERF2.11* increased 2 folds in the adventitious roots and 2.2 folds in the seminal roots, while the same gene in TF57 only increased 1.2 folds in the adventitious roots and 1.3 folds in the seminal roots. In the spike tissue *HvERF2.11* was expressed oppositely between TF57 and TF58 under the waterlogging condition. While the level of *HvERF2.11* increased in TF57, it decreased in TF58. However, the absolute level in the spike tissue was higher in TF58 than in TF57. Since the expression level of *HvERF2.11* was much higher in TF58 than in TF57 under the waterlogging condition, this, combined with the fact that TF58 is tolerant and TF57 is intolerant to waterlogging stress, suggests that the level of *HvERF2.11* might be a key factor in determining waterlogging tolerance. However, it cannot be ruled out the possibility that the existing SNPs between TF57 and TF58 might also be a cause to result in different waterlogging tolerance.

### 2.3. Overexpression of HvERF2.11 in Arabidopsis Enhances Plant Waterlogging Tolerance

To see if *HvERF2.11* was responsible for waterlogging tolerance, transgenic *Arabidopsis* plants overexpressing the *HvERF2.11* gene from TF58 were generated. Transgenic plants that expressed *HvERF2.11* were confirmed by RT-PCR (Figure 5A). Five week-old plants from three homozygous transgenic lines and five week-old WT plants were subjected to waterlogging stress for 2 weeks. As can be found in Figure 5B, under the normal condition, there was no difference in growth or morphological phenotypes between the transgenic lines and WT. Either plant height (Figure 5C), SPAD value (Figure 5D), shoot dry weight (Figure 5F) or root length (Figure 5G) are all similar between the transgenic lines and WT. However, under the waterlogging treatment, differences between the transgenic lines and WT were observed. Firstly, the transgenic lines obviously grew better than the WT plants. Secondly, either plant height (Figure 5C), SPAD value (Figure 5D), shoot fresh weight (Figure 5E), shoot dry weight (Figure 5F) or root length (Figure 5G) were all much bigger in the transgenic lines than in the WT plants. Finally, the transgenic lines were more survival than in the WT plants (Figure 5H). In more details, the heights of the transgenic lines were only reduced by 30.5, 27.4 and 32.5%, respectively, whereas the heights of the WT plants were reduced by 49.1% on the average (Figure 5C). The SPAD value was 61.6% lower in the WT, and this was only 29.5, 35.3 and 31.4% lower in the three tested transgenic lines (Figure 5D). The shoot fresh weights of the transgenic lines were 50.0, 19.7 and 17.9%, respectively, which were slightly lower than those of the WT plants (Figure 5E), but the shoot dry weight in the transgenic lines only decreased by 40.5, 16.1 and 46.3%, while it decreased by 51.0% in the WT plants (Figure 5F). In addition, the root lengths of the WT plants reduced greater than that of the transgenic lines during waterlogging stress (Figure 5G). Furthermore, the average survival rate of the transgenic lines after waterlogging was 69.8% in contrast to the rate of 27.6% in the WT plants (Figure 5H). Taken together, these data indicate that the overexpression of *HvERF2.11* in *Arabidopsis* significantly enhances plant waterlogging tolerance.

### 2.4. Overexpressing of HvERF2.11 Increased the Activities of Antioxidant Enzymes

To examine physiological responses of the transgenic plants vs the WT plants under both normal and waterlogging conditions, a time course of waterlogging treatment was applied to measure the activities of antioxidant enzymes (SOD, CAT, and POD), ADH activity and proline content between the transgenic plants and the WT plants. As shown in Figure 6, no significant physiological difference was observed between the transgenic plants and the WT plants before waterlogging treatment. Either the activities of antioxidant enzymes (SOD, CAT, and POD), ADH activity or proline content were all in low levels from each of the WT plants and the transgenic plants. However, when the plants were treated by waterlogging, the activities of antioxidant enzymes (SOD, CAT, and POD) and ADH activity increased markedly over the time of treatment in both WT and transgenic plants. It seemed that at day 6 of treatment, the activities of SOD, CAT, POD and ADH reached a peak and then dropped at day 9 of treatment except for SOD in the transgenic plants, which maintained its activity. In more details, at day 6 of waterlogging, the SOD activity increased by 24.9% in the WT plants, while in the three transgenic lines this activity increased by 54.6, 48.3 and 45.3%, respectively (Figure 6A), so did the POD activity which increased by 48.8% in the WT plants and by 64.4, 65.3 and 69.9%, respectively, in the three transgenic lines (Figure 6C). In terms of the CAT activity, it increased by 1.5-fold in the WT plants, while this increased by 2.2, 2.1 and 2.1-fold, respectively, in the three transgenic lines (Figure 6B). Similarly, the ADH activity increased by 1.3-fold in the WT plants and in the three transgenic lines the same activity increased by 2.1, 2.3 and 1.9-fold, respectively (Figure 6D).

Proline, which is an important osmotic adjustment substance for stress tolerance, had different contents during the course of waterlogging treatment (Figure 6E). Before the waterlogging treatment, proline contents were similar between the transgenic lines and the WT plants. However, after the waterlogging treatment, proline content in the WT plants increased sharply. It reached a peak at day 6 of treatment and then dropped at day 9 of treatment. By contrast, the same proline content in the transgenic lines was almost unchanged at day 3 of treatment. Although it increased at day 6 of treatment, but not significantly. Different from the activities of SOD, CAT, POD and ADH which dropped at day 9 of waterlogging treatment, proline content continued to increase at this day. It was calculated that at day 9 of treatment, the proline content increased by 2.2, 2.3 and 2.1-fold, respectively, in the three transgenic lines (Figure 6E). Despite of its increasing, the absolute proline content in the transgenic lines was lower than that in the WT plants at day 9 of treatment. Now it can be concluded that the overexpression of *HvERF2.11* enhanced the efficiency of antioxidant systems, which in turn imparts waterlogging stress tolerance.

### 2.5. Overexpression of HvERF2.11 Increases the Expression of Stress-related Genes

To understand the molecular mechanism involved in the *HvERF2.11* response to the waterlogging stress, six stress-related genes (*AtSOD1*, *AtCAT1*, *AtPOD1*, *AtADH1*, *AtPDC1* and *AtACO1*) were examined for their expression levels in the transgenic plants over-expressing *HvERF2.11* (Figure 7). It appeared that the expression patterns from *AtSOD1*, *AtPOD1* and *AtACO1* were very similar to each other, while the expression patterns from *AtCAT1*, *AtADH1* and *AtPDC1* were comparable. The expression of *HvERF2.11* was significantly increased in transgenic lines under waterlogging stress. For *AtSOD1*, *AtPOD1* and *AtACO1*, all of these genes in either the transgenic plants or the WT plants reached a peak level at day 3 of waterlogging treatment and then dropped over the time of treatment. At day 9 of treatment, the expression level of *AtSOD1* in the transgenic plants and the WT plants became the lowest among the whole course of treatment. In contrast, *AtCAT1*, *AtADH1* and *AtPDC1* reached a peak level at day 6 of waterlogging treatment. Although the levels dropped at day 9 of treatment, they were higher than their corresponding ones in both the transgenic plants and the WT plants before the waterlogging treatment. Nevertheless, the above results indicate that overexpression of *HvERF2.11* increases the expression of stress-related genes.

## 3. Discussion

AP2/ERF is one of the largest plant TF super families which plays an important role in biotic and abiotic stresses. In our previous study, we identified 121 HvAP2/ERF genes in barley and in this study we identified a novel ERF gene named *HvERF2.11* in barley. Although the total number of AP2/ERF genes identified in barley is smaller than that identified in rice, which is 174, or in *Arabidopsis*, which is 148, the domains contained such as AP2, RAV, DREB and ERF in these genes are common among these plant species. However, intriguingly, our phylogenetic analysis showed that barley AP2/ERF genes are closer to those from *Arabidopsis* than those from rice. This probably indicates that barley and *Arabidopsis* had a common ancestor for AP2/ERF genes and that AP2/ERF genes existed in *Arabidopsis* prior to the separation of the monocot/dicot groups. After the separation, AP2/ERF genes were evoluted into monocot and dicot ancestors. This has been evidenced that many *Arabidopsis* genes were conserved in other species including monocots.

*HvERF2.11* from TF58, which is waterlogging tolerant, was found to have 11 nucleotides different from that from TF57, which waterlogging intolerant. These single nucleotide changes results in six amino acid changes. In the case of TF57, one amino acid change occurred in the AP2 domain. Previous study showed that in the genes of the DREB subfamily, amino acid residues of Val-14 and Ala-14, respectively, were considered to be important in the gene functions [15]. Thus it cannot be ruled out that the difference in the waterlogging tolerance between TF57 and TF58 might due to the single nucleotide polymorphisms or especially due to the amino acid change in the conserved AP2 domain in TF57. To determine this point, further experimental conformation of roles of these 11 single nucleotide polymorphisms in waterlogging tolerance is required. Nevertheless, these single nucleotide polymorphisms, or at least some of them if not all, could be used as a marker to distinguish between waterlogging tolerance genotypes and waterlogging intolerance genotypes.

In this study, our data revealed that overexpression of *HvERF2.11* in *Arabidopsis* conferred higher waterlogging-tolerance on the transgenic plants, when compared with WT. The transgenic *Arabidopsis* showed a significantly higher plant height, fresh weight, root length and survival rate than WT plants under waterlogging conditions. Waterlogging has been known to induce adverse effects on several physiological and biochemical processes. The change of antioxidant and fermentative enzymes is one of these processes. We found that all the examined antioxidant enzymes such as SOD, CAT, POD and ADH increased their activities under waterlogging stress. More activities of antioxidant enzymes were accumulated in the transgenic *Arabidopsis* compared to the WT under the waterlogging condition. Correspondingly, the genes encoding these enzymes also increased their expression levels. The results were also observed in *Actinidia deliciosa* that the overexpression of *ADH1*, *ADH2*, *PDC1* and *PDC2* genes in *Arabidopsis* had been shown to enhance waterlogging tolerance [29,30].

The genes encoding the antioxidant enzymes (*AtSOD1*, *AtCAT1*, *AtADH1*, *AtPDC1*) of transgenic *Arabidopsis* did not up-regulated under normal growth conditions, but *AtACO1* expression was increased significantly. This suggests that *HvERF2.11* might directly interacts with the *AtACO1*, which is an important gene used for synthesizing gaseous phytohormone ethylene. Ethylene plays an important role in modifying plant response to oxygen deficiency and inducing the aerenchyma and adventitious roots primordia formation as signal transducers [31]. It is thus concluded that *HvERF2.11* induction is not sufficient for activating the genes encoding the antioxidant enzymes and that hypoxic conditions are required. Accordingly, *HvERF2.11* is constitutively expressed and further up-regulated by ethylene under O_2_ deprivation. It could regulate the expression of the antioxidant enzymes indirectly, thereby enhancing plant waterlogging tolerance.

Proline is an osmolyte used for osmotic adjustment that normally accumulates in large quantities under adverse environments. It also has a role in stabilizing the structures of biomacromolecules, such as membranes and proteins, and serves as a ROS scavenger [32]. The proline content is relevant to drought and salt stress [33]. In this study, the discovery of significantly high proline content in the WT plants compared to that in the transgenic plants under waterlogging. Meanwhile, the proline content all increased in the transgenic lines and WT under waterlogging. This could be explained by that *HvERF2.11* could decreased the expression of genes related to proline synthesis under waterloggign stress. This also indicated that the increased of proline content did not improved the waterlogging tolerance. It is believed that more unknown genes that participated in interaction with *HvERF2.11* which needs further investigations in the future.

In conclusion, *HvERF2.11* is significantly induced under waterlogging stress. Transgenic *Arabidopsis* with *HvERF2.11* enhances plant waterlogging tolerance, which might come from increased ethylene synthesis gene (*AtACO1*), activities of fermentation genes (*AtADH1* and *AtPDC1*) and antioxidant enzyme genes (*AtSOD1*, *AtCAT1* and *AtPOD1*). *HvERF2.11* could be used as a tool to enhance plant waterlogging tolerance or as a marker to distinguish between waterlogging tolerance genotypes and waterlogging intolerance genotypes for future barley breeding.

## 4. Materials and Methods

### 4.1. Plant Materials and Waterlogging Treatment

Two barley genotypes (TF57 and TF58) derived from a DH population (TX9425/Franklin) were used for the analysis of gene expression. TX9425 is a feed barley with waterlogging tolerance and originates from China, while Franklin is an Australian malting barley and is susceptible to waterlogging. TF57 is a waterlogging-sensitive line, while TF58 is a waterlogging-tolerant line [28,34]. The seeds of the two lines were sown in plastic pots (25 cm in height and 22 cm in diameter) filled with a mixture of nutritional substance and vermiculite. Plants were grown in a greenhouse with a temperature of 22 ± 2 °C/day and 18 ± 2 °C/night. Waterlogging treatment was performed with water of 3.0 cm above the nutritional substance surface, started at the booting stage and continued for two weeks. The control plants maintained 60–70% of soil moisture. At 14 days of the waterlogging treatment, leaves, adventitious roots, seminal roots, nodal roots and developing spikes were collected with washing and then immediately frozen in liquid nitrogen for further analysis.

### 4.2. Total RNA Isolation and HvERF2.11 Gene Cloning

Total RNA was extracted from different organs (leaves, adventitious roots, nodal roots, seminal roots and spikes) of barley using the Trizol reagent (Invitrogen, Carlsbad, CA, USA) with RNase-free DNase I (TaKaRa, Japan) that was used for removing genomic DNA. cDNA synthesis was performed using M-MLV reverse transcriptase (TaKaRa, Otsu, Shiga, Japan) according to the manufacturer’s instructions. Amplification of *HvERF2.11* was done using a forward primer (5′- AAAGCTCGCCTCCTCCATAC-3′) and a reverse prime (5′-CAGGCATATGACCCAAGGTG- 3′).

### 4.3. Sequence Analysis of HvERF2.11 Gene

The obtained full-length sequence of *HvERF2.11* was analyzed for its amino acid composition with DNAMAN software (Lynnon Biosoft, USA) and for its molecular weight and pI with the online software of ExPASy ProtParam (http://web.expasy.org/protparam/). Homologs of *HvERF2.11* in other plant species were analyzed by the BLAST tool of NCBI (https://blast.ncbi.nlm.nih.gov/Blast.cgi). The phylogenetic tree was constructed by MEGA 6.0 program using the neighbor-joining method and 1000 bootstrap replicates (http://www.megasoftware.net) [35].

### 4.4. Gene Expression Analysis Using qRT-PCR

Primers used for qRT-PCR analyses were designed using the Primer Premier 5.0 (Premier Biosoft International, Palo Alto, CA, USA). All the primers are listed in Appendix A. The 20-µL reaction for qRT-PCR contained 1µL cDNA, 10 mM Tris-HCl (pH 8.5), 50 mM KCl, 2 mM MgCl_2_, 0.4 µL DMSO, 200 mM dNTPs, 10 pmol/µL specific PCR primers, 1 U Taq DNA polymerase, 0.5 µL SYBR GREEN I fluorescence dye. qRT-PCR was carried out in an Applied Biosystems ViiA^TM^7 Real-Time PCR System (Carlsbad City, CA, USA) under the following condition: 94 °C for 5 min, followed by 40 cycles at 94 °C for 30 s, 58 °C for 30 s, 72 °C for 30 s, and a final extension of 72 °C for 5 min. *Hvactin* and *Atactin* were used as an internal control. Ct values were determined by the Applied Biosystems ViiA^TM^ 7 software with default settings. Differences between the Ct values of target gene and *Hvactin* was calculated as △Ct = Ct _target gene_ − Ct _Actin_, and the relative expression levels of target genes were determined as 2^−△Ct^. The same analysis on the same sample was repeated three times. Three biological replicates were also applied. The average values of 2^−△Ct^ were used to determine difference in gene expression. Statistical significance was tested using the Student’s *t*-test (*p* < 0.05) [28].

### 4.5. Plant Transformation of HvERF2.11

The coding sequence of the *HvERF2.11* gene from TF58 was introduced into the pDONR221 vector using the Gateway BP Clonase enzyme mix (Invitrogen, USA) with a primer pair, Gate-25-F and Gate-25-R (Appendix A). After that, this coding sequence was further transferred into the pB2GW7 vector using the GATEWAY cloning technology (Invitrogen, USA).

Transformation of the recombinant plasmid containing the *HvERF2.11* gene into *Agrobacterium tumefaciens* strain GV3101 was carried out according to Bechtold and Pelletier [36]. Further transformation into *Arabidopsis* (Columbia) was as previously described by Clough and Bent [37]. Transgenic *Arabidopsis* lines were selected according to seedling growth on hygromycin-containing (30 mg/mL) MS medium. After two weeks on selection medium, green seedlings (T1 plants) were transferred to soil pots and grown to maturity in a growth room. The PCR-positive plants as transgenes were grown to maturity and seeds were collected (T2 seed). T2 seeds were germinated on hygromycin selection medium again and the one-copy lines were identified by examining the segregation ratio (3:1) of the hygromycin selectable marker. Each one-copy line was maintained growth to set seeds until T3 generation. Homozygosity was obtained in the third (T3) generation and subsequently used for downstream analyses.

### 4.6. Analysis of Transgenic Lines for Waterlogging Tolerance

*Arabidopsis* seedlings were grown in 1/2 MS medium containing 3% sucrose and 0.8% (*w/v*) agar in a chamber at 19–21°C, with a photoperiod of 16/8 h (day/night) for 10 days, then transplanted into nutrient soil. T3 lines of the transgenic *Arabidopsis* (five week-old) were used for the waterlogging treatment together with wild-type (WT) seedlings. The pots were placed in a plastic container and then filled with tap water to 3.0 cm above the soil surface. The control plants were subjected to normal irrigation (60–70% soil moisture content). After the treatment of two weeks, the plant height, root length, shoot fresh weight, dry weight and survival rate were measured. If the yellow leaves more than 80% in the plant, the plants were decided to be dead. This experiment was independently repeated three times. Statistically significant differences were detected with SPSS version 17.0 statistical software (SPSS Corp. Chicago, IL, USA).

### 4.7. Detection of the Physiological Parameters Involved in Waterlogging Tolerance

To examine physiological parameters of transgenic lines overexpressing *HvERF2.11* as well as WT, five week-old soil-grown plants were treated with waterlogging for 3 days, 6 days, 9 days, respectively, and physiological parameters were measured as follows. Chlorophyll content was measured by a soil plant analysis development (SPAD) meter (SPAD-502Plus, Konica Minalta, Osaka, Japan).

Fresh leaves (0.5 g each) were rinsed thoroughly with distilled water. The crude enzymatic extracts of each genotype were prepared in 0.05 M phosphate buffer (pH 7.8) after being ground with a pestle and being milled to powder in liquid nitrogen. The homogenate was filtered through four layers of muslin cloth and centrifuged at 12 000 g for 10 min at 4 °C. The final supernatants were used for physiological and biochemical assays. The superoxide dismutase (SOD) activity was measured using the xanthine oxidase (hydroxylamine) method. The peroxidase (POD) activity was measured using guaiacol as the substrate. The catalase (CAT) activity was assayed by measuring the rate of decomposition of hydrogen peroxide (H_2_O_2_) at 240 nm. Alcohol dehydrogenase (ADH) was measured using acetaldehyde colorimetry. Antioxidase activities, alcohol dehydrogenase (ADH) and proline content were measured using the corresponding assay kits from Jiancheng Bioengineering Institute at Nanjing according to the manufacturer’s instructions.

### 4.8. Expression Analysis of Waterlogging-related Genes in Transgenic Lines

Total RNAs were extracted from the shoots of WT and the transgenic lines grown at the normal irrigation and waterlogging conditions, respectively. After cDNA synthesis, the expression of 6 waterlogging-related genes (*AtADH1*, *AtPDC1*, *AtSOD1*, *AtPOD1*, *AtCAT1*, *AtACO1* and *HvERF2.11*) were analyzed by qRT-PCR. As before, *Arabidopsis actin* was used as an internal control. All the primers used in this experiment are listed in Appendix A. Three biological replicates and three technical replicates were performed for each sample.

## Figures and Tables

**Figure 1 ijms-21-01982-f001:**
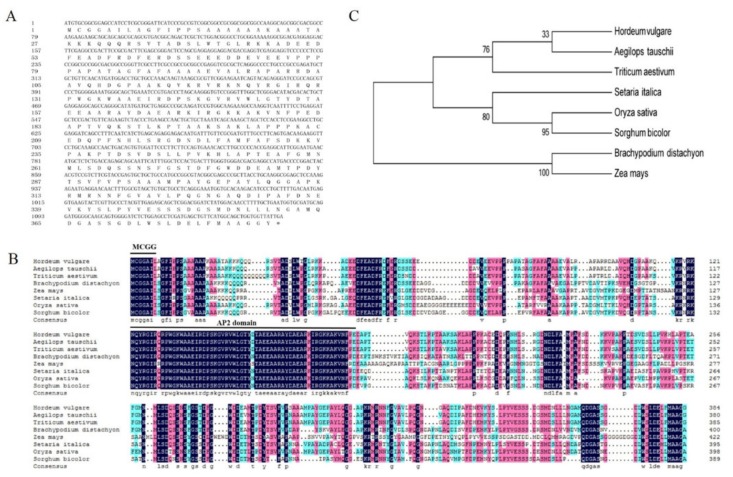
Sequence alignment and phylogenetic analysis of *HvERF2.11* proteins from barley and other plant species.(**A**) Nucleotide and predicted protein sequence of *HvERF2.11*. (**B**) Multiple sequence alignment of *HvERF2.11* with homologous proteins from other species. (**C**) Phylogenetic tree analysis based on the eight ERFs protein sequences *Hordeum vulgare* (BAK03679.1), *Aegilops tauschii* (XP 020194933.1), *Triticum aestivum* (AFP49824.1), *Setaria italic* (XP 004956913.1), *Oryza sativa* (AAV98700.1), *Sorghum bicolor* (XP 021309361.1), *Brachypodium distachyon* (XP 00357816.1) and *Zea mays* (NP 001149434.1). The phylogenetic tree was constructed by the neighbor-joining method. The numbers on the nodes indicate bootstrap values from 1000 replicates.

**Figure 2 ijms-21-01982-f002:**
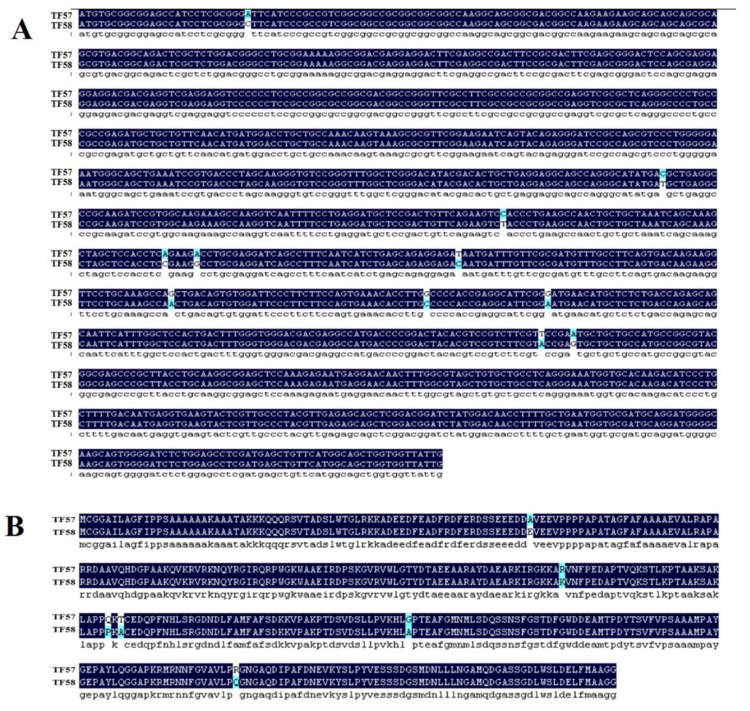
CDS and amino acid sequence comparison of HvERF2.11 gene between TF57 and TF58. (**A**) CDS sequence comparison of HvERF2.11 gene between TF57 and TF58. (**B**) Amino acid sequence comparison of HvERF2.11 gene between TF57 and TF58.

**Figure 3 ijms-21-01982-f003:**
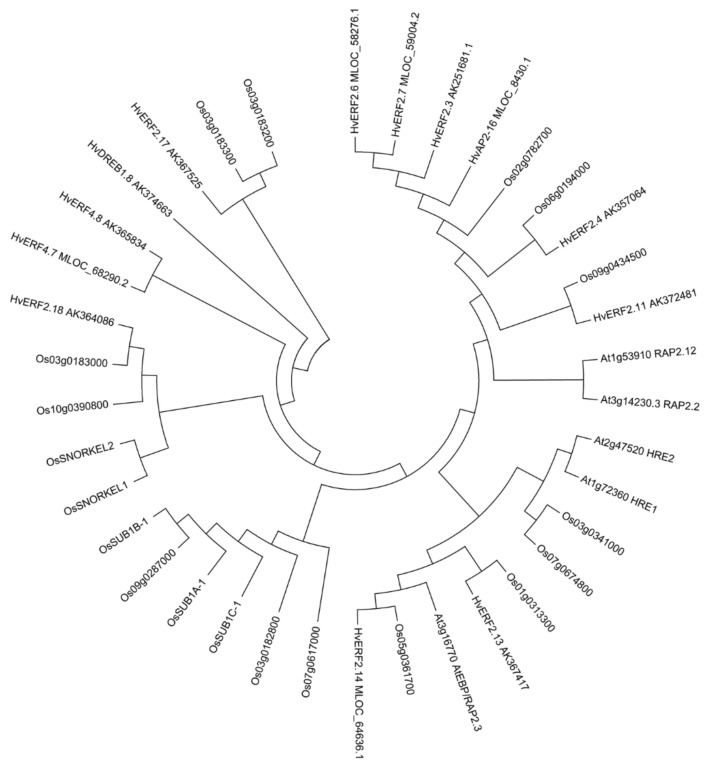
Phylogenetic tree of ERF TFs with conserved N-terminal motif (MCGGAI/L) in Arabidopsis, rice and barley. The phylogenetic tree was constructed by the neighbor-joining method. The numbers on the nodes indicate bootstrap values from 1000 replicates.

**Figure 4 ijms-21-01982-f004:**
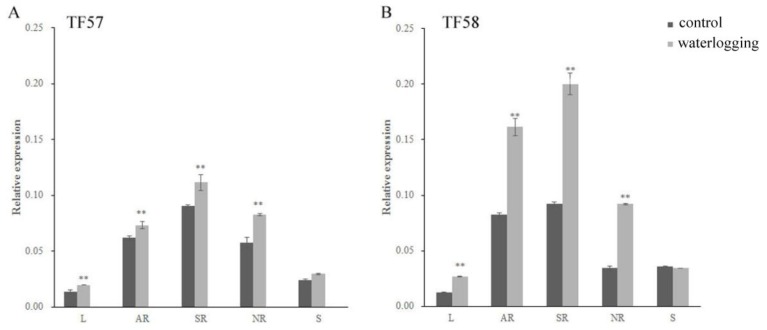
Expression levels of *HvERF2.11* gene in TF57 and TF58, RT-PCR analysis of transgenic and wild-type plants. (**A**) Expression levels of the *HvERF2.11* gene under waterlogging stress in different organs of TF57. Results are the mean ± SD. * and ** represent the significant differences at *p* < 0.05 and *p* < 0.01, respectively. (**B**) Expression levels of the *HvERF2.11* gene under waterlogging stress in different organs of TF58. The mean value and standard deviation were obtained from three independent experiments. The data represent mean ± SD of three biological repeats with three measurements per sample. * and ** represent the significant differences at *p* < 0.05 and *p* < 0.01, respectively. L: leaf; AR: adventitious root; SR: seminal root; NR: nodal root; S: spike.

**Figure 5 ijms-21-01982-f005:**
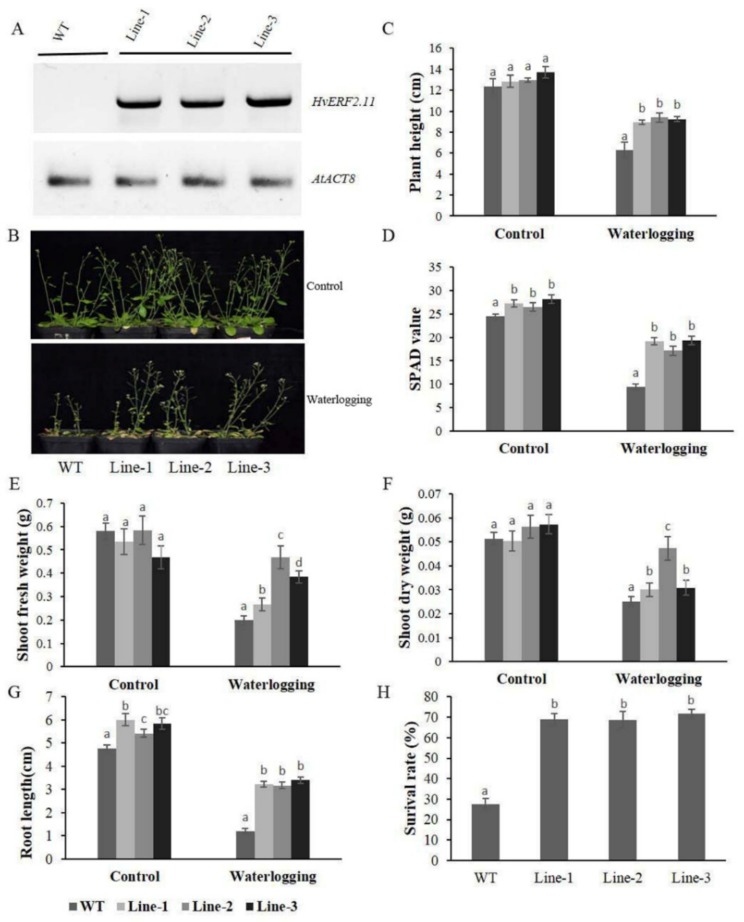
Waterlogging tolerance assay of *HvERF2.11* overexpression lines (Line1, Line2, Line 3) and wild-type (WT). (**A**) RT-PCR analysis of transgenic and wild-type plants. *AtACT8* was chosen as control gene. (**B**) Five week-old plants were subjected to waterlogging stress for further 2 weeks. (**C**) Plant height. (**D**) Soil-plant analysis development (SPAD) value (based on chlorophyll meter reading). (**E**) Shoot fresh weight. (**F**) Shoot dry weight. (**G**) Root length. (**H**) Surival rate in the wild-type and *HvERF2.11* transgenic lines were measured under control and waterlogging stress. Values are the means ± SD. Means were generated from three independent measurements. Data were analyzed by one-way analysis of variance followed by Duncan’s test. Different letters represent statistically significant differences (*p* < 0.05).

**Figure 6 ijms-21-01982-f006:**
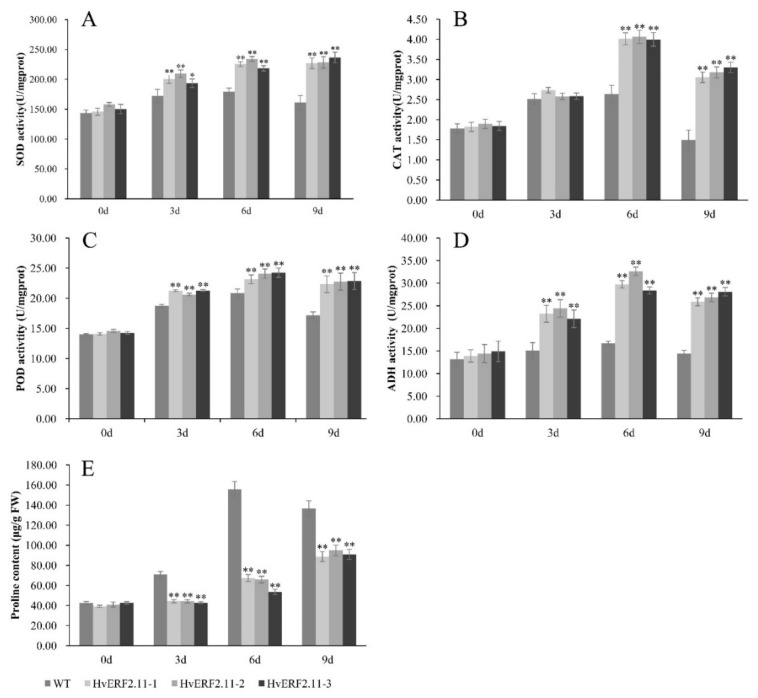
Analysis of SOD, CAT, POD, ADH activities and proline content were carried out in transgenic lines and WT under waterlogging stressed conditions. SOD, CAT, POD, ADH and proline levels. (**A–E**) were measured in the leaves of plants subjected to waterlogging stress 3 days, 6 days, 9 days. The mean value and standard deviation were obtained from three independent experiments. The data represent mean ± SD of three biological repeats with three measurements per sample. Asterisks indicate significant differences from WT as determined using Student’s *t*-test (* *p* < 0.05; ** *p* < 0.01).

**Figure 7 ijms-21-01982-f007:**
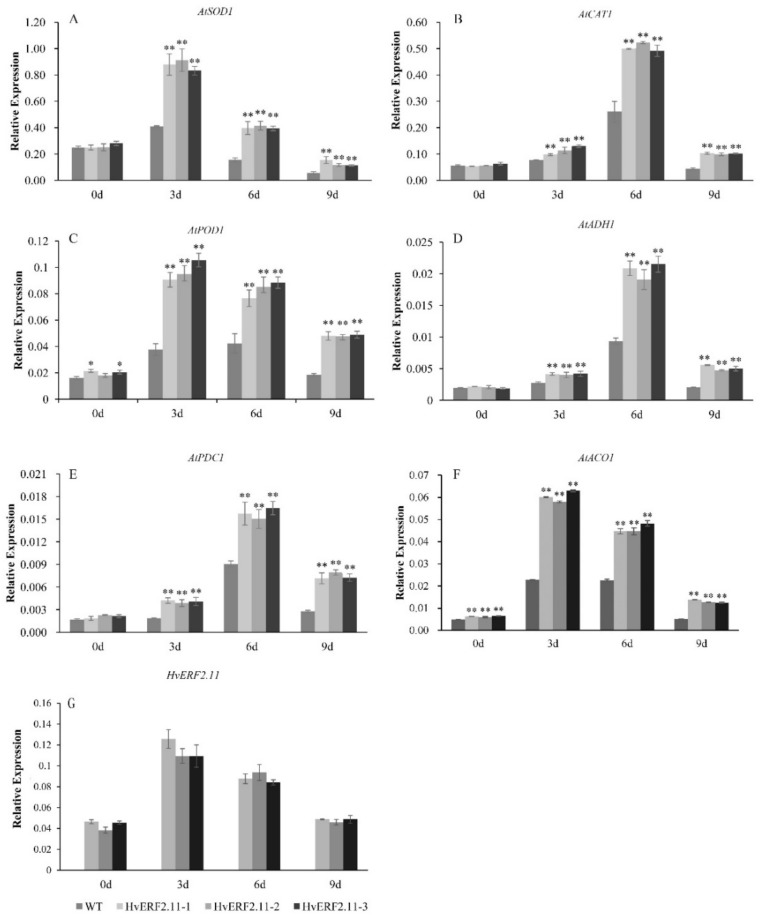
Expression analysis of stress-responsive genes in *HvERF2.11* overexpression lines and WT under waterlogging stresses. The relative expression levels of stress-responsive genes (*AtSOD1*, *AtCAT1*, *AtPOD1*, *AtADH1*, *AtPDC1*, *AtACO1*) and *HvERF2.11* were determined by qRT-PCR (**A**–**G**). After 3 days, 6 days, 9 days waterlogging treatments, respectively. Seedlings harvested before treatment were used as control. Relative expression levels of these six genes were normalized to the transcripts of *AtActin* in the same samples. The mean value and standard deviation were obtained from three independent experiments. The data represent mean ± SD of three biological repeats with three measurements per sample. Asterisks indicate significant differences between transgenic plants and WT according to Student’s t-test (* *p* < 0.05; ** *p* < 0.01).

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
