# Peer review of "Overexpression of Barley Transcription Factor *HvERF2.11* in *Arabidopsis* Enhances Plant Waterlogging Tolerance"

_ijms, 2020, doi:10.3390/ijms21061982_

Round 1

Reviewer 1 Report

The manuscript by Luan et al. deals with the characterization of the transcription factor (TF) HvERF2.11 from barley (Hordeum vulgare) as enhancer of waterlogging tolerance. Arabidopsis plants overexpressing HvERF2.11 showed significantly higher tolerance to waterlogging stress than wild-type plants, in terms of growth and survival; biochemical analyses of antioxidant enzymatic activities and proline content also indicated in the transgenic plants a more effective defense system as well as enhanced expression of stress-related genes.

The subject of TFs involved in waterlogging tolerance in crops is quite relevant and the results included in this manuscript can add new knowledge to the topics. Nevertheless, the manuscript should be improved to help readers reaching conclusions by means of well described and commented data.

Some major concerns:

English should be generally revised (i.e. see sentences L. 43-44; 57-60 ..)

“Reactive oxygen species” are mentioned as measured in transgenic plants at the end of Abstract and Introduction but they were not reported elsewhere in the manuscript. 

Discussion should be carefully revised to address the main goals reached by the presented results. Comments are missing as for the characterization of Arabidopsis transgenic plants compared to wild-type plants. For instance, while commenting antioxidant enzyme results, increases detected in transgenic plants were poorly discussed.The authors should better explain why proline behaviour suggests that “it is unrelated to HvERF2.11” when transgenic plants showed a considerably lower proline increase.

Materials and methods should include more details i.e. regarding the Arabidopsis transformation protocol (Paragraph 4.5.) or enzyme activity measurements (Par. 4.7.) .

Minor hints:

Graphs contain too small characters thus hardly readable (see Fig. 4).  In Fig.5D, SPAD value should be explained.

L.203: Figure 7 should be 6.

Author Response

Dear Editors and Reviewers: 

Thank you very much for sending us the reviewers’ comments on our manuscript (Manuscript ID: ijms-727426) entitled “Overexpression of Barley Transcription Factor HvERF2.11 in Arabidopsis Enhances Plant Waterlogging Tolerance”. We sincerely thank the reviewers for their valuable comments and criticisms. We have addressed their comments one by one and have made correction accordingly as indicated by tracked changes in the revised version. Below are our responses to the reviewer’s comments.

Response to reviewer 1:

Comment 1: English should be generally revised (i.e. see sentence+es L. 43-44; 57-60.)

Response 1: The paper has been reedited by a native English speaker.

For example, your concerned sentences in L43-44 ‘The significant change in gene expression is transcription factors (TFs), of which many were found to be waterlogging-regulated [11-13].’ has been changed to ‘Many stress-associated transcription factors (TFs) were also found to vary in expression level under waterlogging stress [11-13]. For example, ERFs and WRKYs were upregulated, while DNA-binding with one finger (Dof) TFs were down-regulated under waterlogging stress [13] ’.

and in L57-60 ‘Transgenic plants with HRE1 and HRE2 increase levels of the genes related to anaerobic and ethanolic fermentation such as PDC1 and ADH1, thereby improving low-oxygen stress tolerance [20].’ has been changed to

HRE1 and HRE2 improved the tolerance of low-oxygen stress by up-regulated genes related to anaerobic and ethanolic fermentation such as PDC1 and ADH1 [20].’

Comment 2: “Reactive oxygen species” are mentioned as measured in transgenic plants at the end of Abstract and Introduction but they were not reported elsewhere in the manuscript. 

Response 2: Thanks for this point. ROS have been removed from both Abstract and Introduction in the revised version.

Comment 3: Discussion should be carefully revised to address the main goals reached by the presented results. Comments are missing as for the characterization of Arabidopsis transgenic plants compared to wild-type plants. For instance, while commenting antioxidant enzyme results, increases detected in transgenic plants were poorly discussed. The authors should better explain why proline behaviour suggests that “it is unrelated to HvERF2.11” when transgenic plants showed a considerably lower proline increase.

Response 3: Thanks for the advice. We have largely modified the discussion according to your suggestions, which can be easily identified by track change in the revised version.

Comment 4: Materials and methods should include more details i.e. regarding the Arabidopsis transformation protocol (Paragraph 4.5.) or enzyme activity measurements (Par. 4.7.)

Response 4: Thanks for this comment, more details have been given in the revised manuscript, which can be found by track change

Comment 5: Minor hints: Graphs contain too small characters thus hardly readable (see Fig. 4).  In Fig.5D, SPAD value should be explained.

L.203: Figure 7 should be 6.

Response 5: Thanks the reviewer for this comment. The small characters have been changed in Fig. 4, while SPAD value has been explained as ‘Soil-plant analysis development (SPAD) value (based on chlorophyll meter reading)’ in Fig.5D

L.203: Figure 7 has been changed to 6 now.

Finally, we sincerely thank the reviewers again for their constructive comments or suggestions, which help to improve the quality of this manuscript.

Best wishes,

Dr. Rugen Xu

Reviewer 2 Report

This manuscript describes about a barley ERF transcription factor that has a potential to regulate waterlogging tolerance in Barley. The authors had previously identified a barley ERF gene that is upregulated in a waterlogging stress. In this study, the authors analyzed phylogenetic relationship of the gene with ERF genes in other plant species. The authors also produced the ERF gene overexpressed Arabidopsis and found that the gene conferred waterlogging tolerance on the transgenic Arabidopsis.  The results are interesting but it is necessary to clarify that the shown data are reliable. The authors used qRT-PCR to detect the expression levels of HvERF2.11 in barley tissues (Fig. 4) and genes related to waterlogging tolerance in transgenic Arabidopsis (Fig. 7). In these experiments, the number of the experimental replications are not indicated (Fig. 4) or insufficient (Fig. 7). In papers published in standard level journals, this kind of experiments requires at least three “biological” replicates, which means that RNA samples are prepared from three independent samples and examined independently, as indicated by their own results of enzyme activity assays shown in Fig. 6. Mean values from the three independent samples should be indicated. Mean values of three independent assays shown in Fig. 7, however, are meaningless. The authors should show results of the experiments with RNAs that are prepared from three independent samples. If they cannot prepare, these results are not acceptable as main text figures to indicate evidences of their claims. However, if the journal policy allows (I think it depends on the editor), these results may be indicated as supplemental data and the author may be allowed to claim the possibilities as their speculation based on the results under insufficient conditions.

Other points

  1. In Abstract, the author claimed that ROS reduced their accumulation (Line 23), but there are no direct experimental evidences on the claim in the manuscript.
  2. In line 39, ACS and ACO are not involved in glycolysis pathway or sucrose metabolism but in ethylene production. Invertase (line 41) is involved in sucrose metabolism.
  3. In line 45, DNA-binding with one finger was … -> DNA-binding with one finger (Dof) TFs were …
  4. It is difficult to understand how the two DH lines were generated. The authors should explain briefly the origin and the properties of the parents of the DH lines.
  5. In Fig. 1C, the accession #s of each ERF should be indicated.
  6. Based on the result of Fig. 3, I think OsSUB1B-1, OsSUB1A-1 and OsSUB1C-1 form a cluster with Os09g0287000, but the description in line 131 is not so. It should be clarified.
  7. Based on the Fig.4, the expression of HvERF2.11 in T57 under a waterlogging condition is increased by a level with significant difference (P<0.01), but the description about it in the text (line 147) is not true.
  8. It is obscure the term “survival rate” in Fig. 5H. The authors should explain when and how the plants were decided to be dead.
  9. Which promoter was used to drive HvERF2.11 in transgenic Arabidopsis? Does it act constitutively? If a constitutive expression promoter was used, and if HvERF2.11 induce directly the expression of the waterlogging tolerance related genes, the tolerance genes should be up-regulated even before the waterlogging stress. However, the results are unlikely so. HvERF2.11 may have an additive effect for an inducer of these tolerance related genes. The authors should discuss about the phenomena. Also, the expression of HvERF2.11 during waterlogging conditions in the transgenic lines should be indicated.
  10. In line 260-, by the delta-delta method to analyze qRT-PCR data, expression levels between different genes cannot be compared.

Author Response

Response to reviewer 2:

Comment 1: The problem about the experimental replicates

Comment 2: Thanks for this comment. The mean values described in the paper came from three independent experiments. This has been indicated in the “Materials and methods”. Now this kind of information is given in figure legends in the revised version.

Comment 2: In Abstract, the author claimed that ROS reduced their accumulation (Line 23), but there are no direct experimental evidences on the claim in the manuscript.

Response 2: Thanks for this point. ROS have been removed from both Abstract and Introduction in the revised version.

Comment 3: In line 39, ACS and ACO are not involved in glycolysis pathway or sucrose metabolism but in ethylene production. Invertase (line 41) is involved in sucrose metabolism.

Response 3: Thanks for this comment. We have modified the original sentence into “pyruvate decarboxylase (PDC) gene and alcohol dehydrogenase (ADH)” in the revised manuscript.

Comment 4: In line 45, DNA-binding with one finger was … -> DNA-binding with one finger (Dof) TFs were …

Response 4: We have changed it in revised manuscript.

Comment 5: It is difficult to understand how the two DH lines were generated. The authors should explain briefly the origin and the properties of the parents of the DH lines.

Response 5: Thanks for this comment. We have given the origin and the properties of the parents of the DH lines in the ‘Materials and Methods’ in the revised version.

Comment 6: In Fig. 1C, the accession #s of each ERF should be indicated.

Response 6: Thanks for this comment. The accession #s of each ERF has been indicated in the revised version, for example, ERF from Hordeum vulgare (BAK03679.1), from Aegilops tauschii (XP 020194933.1), from Triticum aestivum (AFP49824.1), from Setaria italic (XP 004956913.1), from Oryza sativa (AAV98700.1), from Sorghum bicolor (XP 021309361.1), from Brachypodium distachyon (XP 00357816.1) and from Zea mays (NP 001149434.1).

Comment 7: Based on the result of Fig. 3, I think OsSUB1B-1, OsSUB1A-1 and OsSUB1C-1 form a cluster with Os09g0287000, but the description in line 131 is not so. It should be clarified.

Response 7: We have changed it in revised manuscript.

Comment 8: Based on the Fig.4, the expression of HvERF2.11 in T57 under a waterlogging condition is increased by a level with significant difference (P<0.01), but the description about it in the text (line 147) is not true.

Response 8: We have changed ‘Under the waterlogging condition, levels of HvERF2.11 in T57 increased in all the tissues, but not significantly. In contrast, levels of HvERF2.11 in T58 increased significantly in most tissues.’ to ‘HvERF2.11 showed a differential inducible pattern between TF57 and TF58. We found that HvERF2.11 in adventitious roots, seminal roots and nodal roots expressed at a higher level in TF58 than in TF57 under waterlogging stress.’ in the revised version.

Comment 9: It is obscure the term “survival rate” in Fig. 5H. The authors should explain when and how the plants were decided to be dead.

Response 9: We have given how “survival rate” is defined in the revised version. If the yellow leaves more than 80% in the plant, the plants would be decided to be dead.’

Comment 10: Which promoter was used to drive HvERF2.11 in transgenic Arabidopsis? Does it act constitutively? If a constitutive expression promoter was used, and if HvERF2.11 induce directly the expression of the waterlogging tolerance related genes, the tolerance genes should be up-regulated even before the waterlogging stress. However, the results are unlikely so. HvERF2.11 may have an additive effect for an inducer of these tolerance related genes. The authors should discuss about the phenomena. Also, the expression of HvERF2.11 during waterlogging conditions in the transgenic lines should be indicated.

Response 10: Cauliflower mosaic virus (CaMV35S) promoter was used to drive HvERF2.11 in transgenic Arabidopsis. It is a constitutive expression promoter. The tolerance genes did not up-regulate under normal condition, and the reason has been explained in the revised manuscript. The expression of HvERF2.11 during waterlogging conditions in the transgenic lines has been indicated in Figure 7.

Comment 11: In line 260-, by the delta-delta method to analyze qRT-PCR data, expression levels between different genes cannot be compared.

Response 11: Thanks for this comment. The related information has been deleted in the revised manuscript.
